# Chinese Metaphorical Relation Extraction

**Guihua Chen**[1,*] **Tiantian Wu**[1,*]

**Miaomiao Cheng**[1]**, Xu Han**[1]**, Jiefu Gong**[2,3]**, Shijin Wang**[2,3] **, Wei Song**[1]

[1]College of Information Engineering, Capital Normal University, Beijing, China
[2]State Key Laboratory of Cognitive Intelligence, iFLYTEK Research, China
[3]iFLYTEK AI Research (Hebei), Langfang, China
{ghchen, ttwu, miaomiao, hanxu, wsong}@cnu.edu.cn,
{jfgong, sjwang3}@iflytek.com

## Abstract

Metaphors are linguistic expressions that convey non-literal meanings, as well as conceptual mappings that establish connections between distinct domains of experience or knowledge. This paper proposes a novel formulation of metaphor identification as a relation extraction problem. We introduce metaphorical relations as links between two spans in text, a target span and a source-related span. We create a dataset for Chinese metaphorical relation extraction, with more than 4,200 sentences annotated with metaphorical relations, corresponding target/source-related spans, and fine-grained span types. Metaphorical relation extraction is a process that detects metaphorical expressions and builds connections between target and source domains. We develop a span-based end-to-end model for metaphorical relation extraction and demonstrate its effectiveness. We expect that metaphorical relation extraction can serve as a bridge between linguistic metaphor identification and conceptual metaphor identification. Our data and code are available at https://github.com/cnunlp/CMRE.

## 1 Introduction

Metaphor is a pervasive linguistic phenomenon in natural language (Cameron, 2003). It involves expressing non-literal meanings that are not directly derived from the words of the metaphorical expressions and that pose challenges for computational understanding. Metaphor is also a cognitive process (Lakoff and Johnson, 2008). A metaphor consists of 3 parts: the *target*, the *source*, and an *analogical mapping* from the source to the target (Carbonell and Minton, 1983), where the target and the source are concepts or conceptual domains. Metaphor processing has been an important research field in natural language processing.

Metaphor identification is a core task in metaphor processing, which involves recognizing and analyzing metaphorical expressions in text. Metaphorical expressions convey a meaning different from their literal interpretation, such as "*knowledge is power*" and "my car is *drinking* gas". Previous work on metaphor identification has mainly concentrated on two types of expressions.

The first type is syntactic constructions such as noun-is-noun (AisB style), verb-subject, adjective-noun, and verb-direct object constructions (Levin et al., 2014; Tsvetkov et al., 2014). The motivation is to capture the constructions that violate the *selectional preference* (Wilks, 1978). So metaphor identification is formulated as a syntactically-related word-pair classification problem (Tsvetkov et al., 2014; Shutova et al., 2016; Rei et al., 2017). However, this formulation usually requires a parser to extract word pairs that belong to pre-defined checkable relations (Levin et al., 2014), neglecting other types of expressions and ignoring useful contexts in the whole sentence.

The second type is annotating the metaphoricity of individual words in a sentence. The motivation is described by the *metaphor identification procedure* (MIP) (Group, 2007) that a word would be annotated as metaphorical if its meaning in a specific context is different from its basic meaning. Based on this scheme, metaphor identification can be formulated as a sequence labeling task, which has become popular due to the rapid development of deep contextual representation learning (Do Dinh and Gurevych, 2016; Gao et al., 2018) and pretrained language models (PLMs) (Leong et al., 2020). However, this formulation cannot explicitly reflect the conceptual structure of metaphors. For example, in the sentence "my car is *drinking* gas", annotating the verb *drinking* as metaphorical cannot reveal the source-target mapping.

In this paper, we propose a novel way to represent metaphors through metaphorical relations.

---

*Co-first authors.

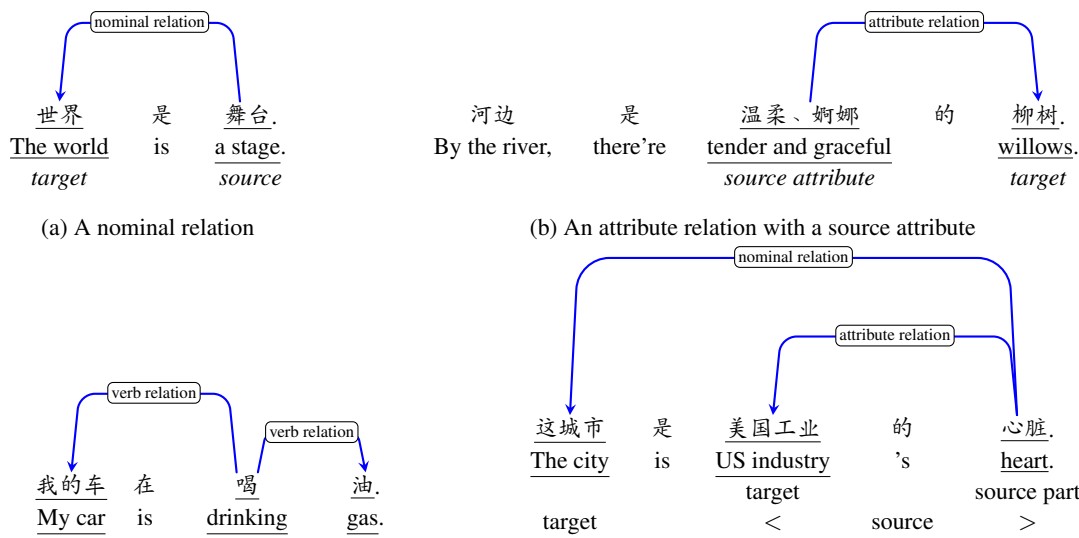

(a) A nominal relation

(b) An attribute relation with a source attribute

(c) A sentence with a simile relation and a verb relation

(d) An attribute relation with a source part and a nominal relation in the same sentence

Figure 1: Examples of metaphorical relations. A metaphorical relation is shown as a link from a source-related span to a target span. Each span region is underlined and has a fine-grained type below. The fine-grained span types characterize the fine-grained metaphorical relation types as labeled on the links.

A metaphorical relation connects two spans, a *target span* and a *source-related span*. We create a metaphorical relation dataset for Mandarin Chinese. The dataset consists of more than 4,200 sentences annotated with metaphorical relations, target/source-related spans, and fine-grained span types. As shown in Figure 1, the fine-grained span types, such as *source, source action, source attribute* and *source part*, can characterize metaphorical relations into the fine-grained types, such as *nominal, verb*, *attribute* and *simile relations*.

Annotating metaphorical relations encompasses the previous two annotation strategies and overcomes their shortcomings. It has the following advantages. First, metaphorical relations use spans instead of words as the basic unit to more precisely capture the characteristics of the target and the source. Second, metaphorical relations build surface connections between the target and the source, overcoming the weakness of annotating metaphorical words only. Although the source is often implicit, the connections can be built based on the actions, attributes, or parts of the source. For example, *my car* and *drinking* form a metaphorical relation, where *my car* is the target and *drinking* is an action of the source. Third, metaphorical relations can cover multiple types of metaphors that have been discussed in previous work, including AisB style, verb, adjective metaphors, and simi-

les since they share a similar underlying cognitive process. So different types of metaphors can be handled in the same framework rather than processed in separate frameworks (Liu et al., 2018; Song et al., 2021b,a).

Based on the annotated dataset, we explore span-based models for metaphor identification. We propose an end-to-end model jointly optimizing the span proposer, fine-grained span type classification, and metaphorical relation extraction. Specifically, we propose a pair-wise span type classification module to predict the types of a pair of spans simultaneously since they are closely associated and support metaphorical relation scoring through the span type features. Experimental results demonstrate the effectiveness of the proposed method compared with the baseline that is directly adapted from an entity relation extraction model. However, there is still room for further improvement in the performance. Our method can be used as a baseline for future research.

To the best of our knowledge, this is the first work that explores span-based relations to annotate and identify metaphors. Metaphorical relations not only reveal the metaphorical expressions but also build surface connections between the target and the source. Because the sources are often implicit in the text for metaphors involving metaphorically used verbs and adjectives, we can further infer com-

plete source-target mappings based on metaphorical relation extraction. So metaphorical relation extraction has the potential to connect linguistic metaphor identification and conceptual metaphor identification (Steen, 1999). We leave this as the future work.

## 2 Related Work

**Metaphor Datasets** The VU Amsterdam Metaphor Corpus (VUA) is the largest and most representative dataset (Steen et al., 2010), annotated based on the *metaphor identification procedure* (MIP) (Group, 2007). The other popular word-level metaphor datasets include MOH-X (Mohammad et al., 2016) and TroFi (Birke and Sarkar, 2006). VUA annotates all individual words, while MOH-X and TroFi annotate verbs and their sizes are relatively small.

Some datasets focus on the metaphoricity of syntactically-related word pairs, together with other attributes such as emotion or novelty (Levin et al., 2014; Tsvetkov et al., 2014; Mohler et al., 2016; Do Dinh et al., 2018).

Shutova and Teufel (2010) built a dataset for annotating source-target domain mappings for metaphorically used verbs, consisting of 761 sentences. The mappings are constructed based on the source and target domain categories borrowed from the Master Metaphor List (Lakoff, 1994).

These datasets are mainly for English. A few datasets cover Spanish, Russian, and Farsi (Mohler et al., 2016; Tsvetkov et al., 2014). Zhang et al. (2018) created a Chinese dataset for analyzing the emotionality of metaphors, which is a relatively large public dataset for Chinese.

Our work differs from previous work in the annotation scheme. We explicitly annotate metaphorical relations between spans related to the target and the source. This way is more general and expressive compared to annotating individual words or syntactically related word pairs.

**Metaphor Identification Methods** Similar to the construction of the datasets, metaphor identification can be viewed as a word-pair classification problem or a word sequence labeling problem. Most of the current models depend on recurrent neural networks (Do Dinh and Gurevych, 2016; Rei et al., 2017; Gao et al., 2018) and pre-trained language models (Leong et al., 2020; Dankers et al., 2020; Su et al., 2020; Li et al., 2021; Ge et al., 2022; Aghazadeh et al., 2022; Li et al., 2023).

Based on the created metaphorical relation dataset, we open a new perspective by exploring the span-based relation extraction models for metaphor identification. Song et al. (2021b) proposed a contextual relation learning method for verb metaphor identification but it is still based on individual word annotations and requires a dependency parser.

**Span-based Entity Relation Extraction** The proposed metaphorical relation extraction model is motivated by the model for coreference resolution (Lee et al., 2017, 2018), which is a span-based relation extraction model. However, it does not explicitly consider span types, which are important in our task to indicate the structure of metaphors. To handle this, we propose to jointly optimize relation extraction and span type classification.

Our model is also similar to the entity relation extraction models. We do not simply adapt the pipeline approaches, e.g., (Zhong and Chen, 2021), or previous joint entity relation extraction models (Dixit and Al-Onaizan, 2019; Eberts and Ulges, 2020) to our task. Instead, we focus on handling the challenge in our task that the span types are hard to determine without considering specific involved metaphorical relations.

## 3 Metaphorical Relation Dataset

In this section, we introduce the created Chinese metaphorical relation dataset. We first introduce metaphorical relations, then describe the details of data annotation, and finally conduct basic data analysis.

### 3.1 Metaphorical Relations

We define a **metaphorical relation** as a link between two spans, a **target span** and a **source-related span**. Considering the example in Figure 1(b), *willows* is the target span, and *tender and graceful* is a source-related span. We use a directed arc from the source-related span to the target span to represent the metaphorical relation. The two spans trigger a metaphor because *tender and graceful* are not usually used to describe *willows*.

A span consists of consecutive tokens to cover precise contextual clues. We expect the span should cover necessary and compact descriptions, which can reflect the properties of the source or the target and provide precise contextual clues. Considering the sentence, 奸诈的威尼斯商人就是一只狡黠的狐狸 (*the treacherous Venetian merchant is a cunning fox*), wider spans with detailed de-

scriptions can provide useful clues for building the mapping between the target and the source.

We define fine-grained types for source-related spans, including:

- **Source**: Concepts that are used as the source of a metaphor. Sources are usually realized in AisB style metaphors while they are often implicit in metaphors involving metaphorically used verbs or adjectives.

- **Source action**: Typical actions or passive actions of an implicit source. Consider an example, "the car is *drinking* gas", where *drink* is an action of humans or animals, indicating a mapping from *animal* to *car*, and *drink* is also a passive action of *water*, resulting in a contrast between *water* and *gas*.

- **Source attribute**: Typical attributes of an implicit source, e.g., the span *tender and graceful* is commonly used to describe *women* which is a potential source concept.

- **Source part**: Parts or related concepts of an implicit source. Figure 1 (d) shows a part relation, where *heart* is part of a person, indicating that the *US industry* is mapped to a person.

With the fine-grained span types, we can categorize some fine-grained metaphorical relations:

- **Nominal relation**: A nominal metaphorical relation consists of a target and a source. The most typical nominal relation is the AisB style metaphors.

- **Verb relation**: A verb metaphorical relation connects a target and a source action.

- **Attribute relation**: An attribute metaphorical relation connects a target and a source attribute or a source part.

- **Simile relation**: We can view simile as a special metaphorical relation type, which also has a target and a source but there exist explicit comparators such as 像*(like)* in context.

In summary, metaphorical relations establish the connection between the source and the target on the surface of the sentence. They do not depend on syntactic patterns and utilize flexible and concise spans to characterize the source and target domains.

## 3.2 Data Annotation

**Data Source** Our annotation is based on the dataset built by Zhang et al. (2018), which is a Chinese dataset for analyzing the emotionality of metaphors. The genres include books, journals, movie scripts, and social media texts. The dataset has 5,494 sentences annotated with metaphorical labels and emotion categories at the sentence level. The metaphorical labels reveal whether a sentence has noun metaphors, verb metaphors, or no metaphors. We annotate metaphorical relations in metaphorical sentences and also include a variety of external literal sentences to make the distribution of metaphorical and literal sentences more balanced.

**Metaphorical Relation Annotation** For each metaphorical sentence, we ask annotators to annotate metaphorical relations, spans, and fine-grained span types. The annotators are two students with a background in linguistics.

We prepare a manual, which describes: (1) the definition of metaphorical relations and the standard to establish a metaphorical relation, which refers to the work of Shutova and Teufel (2010) and Group (2007); (2) the criteria and examples for annotating span boundaries; (3) the definition and examples of the fine-grained span types.

The annotation process is as follows. In the first stage, the annotators would try to annotate part of the metaphorical sentences to understand and follow the manual. We emphasize the importance of consistent and reliable annotation, especially on the span boundaries and the existence of metaphorical relations.

In the second stage, we assign the whole dataset to both annotators and let them annotate independently. We randomly sample about 200 metaphorical sentences that have been re-annotated by them. The rate of inter-annotator agreement on the span pairs is about 77%. There are only a few disagreements due to the span region and most of the disagreements are because some relations are missed by some annotators, especially in sentences with multiple metaphorical relations. For the spans annotated by both annotators, the rate of inter-annotator agreement on fine-grained span types is 93%, which is high, indicating that once people recognize a metaphorical relation, the types of its two spans can be determined easily.

Finally, every sentence has a second pass review to resolve any conflicts and reach an agreement. During annotation, sentences that are too long or

| Dataset | #St | %M | avg. #R |
|---|---|---|---|
| Training | 6,794 | 50 | 1.28 |
| Development | 850 | 50 | 1.27 |
| Test | 850 | 50 | 1.28 |
| Total | 8,494 | 50 | 1.28 |

Table 1: Statistics of the training set, development set, and test set. #St: number of sentences; %M: metaphor percentage; avg. #R: average number of metaphorical relations per metaphorical sentence.

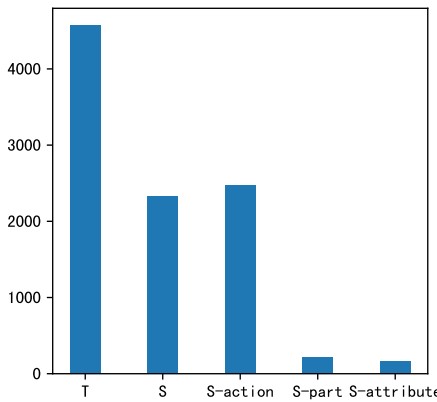

Figure 2: Distribution of the fine-grained span types. T: target, S: source.

have vague meanings should be removed.

**Adding Literal Sentences** We expand external literal sentences to the dataset to keep a balanced ratio between metaphorical and literal sentences. We analyze the initially annotated dataset and alleviate the following artificial bias:

- **Avoiding word bias**: Similes have explicit comparators such as 像 (*like, as*) and AisB style nominal metaphors use 是 (*is, are*). Most sentences in the initial annotated dataset that contain such words are metaphorical. So we should include literal sentences that also contain such words.

- **Avoiding sentence length bias**: We hope to keep the length distribution of literal sentences close to metaphorical sentences to avoid biases in sentence length.

We randomly sample sentences and retrieve sentences with biased words from Baidu Baike, student essays, and books, and manually select literal sentences to have a balanced and unbiased dataset.

Finally, the dataset has 8,494 sentences. The percentage of metaphorical sentences is 0.5. We divide it into training, development, and test sets with a ratio close to 8: 1: 1. The basic statistics are shown in Table 1.

Figure 2 further shows the statistics of the fine-grained span types. Although the imbalanced distribution adheres to our intuition, the number of source attribute/part spans is relatively small. This is a limitation of this work.

## 4 The Proposed Model

### 4.1 Overview

Metaphorical relation extraction aims to extract a set of non-overlapping typed span pairs from sentences and determine the type of these relations.

We propose a span-based end-to-end model. The main architecture is shown in Figure 3. A span proposer would enumerate, score, and select the best spans, which are used for efficiently generating candidate span pairs. Each pair would be fed into two modules: pair-wise span type classification, which is specially designed for determining span types, and metaphorical relation scoring, which estimates the possibility that the candidate pair has a metaphorical relation. The two modules work together to extract metaphorical relations.

### 4.2 Span Proposer

Given a sentence with $M$ tokens $S = w_1, ..., w_M$, we use BERT to get the contextual representation $\mathbf{e}_m$ of each token $w_m$. The span proposer enumerates all possible spans up to a maximum length $L$ in a sentence. For a given span $i$ with the start and end indices $\text{START}(i)$ and $\text{END}(i)$, its representation is computed as follows,

$$\mathbf{g}(i) = [\mathbf{e}_{\text{START}(i)}, \mathbf{e}_{\text{END}(i)}, \overline{\mathbf{e}}_i, \phi(i)], \quad (1)$$

where $\mathbf{e}_{\text{START}(i)}$ and $\mathbf{e}_{\text{END}(i)}$ are the contextual representations of the start and end tokens; $\overline{\mathbf{e}}_i$ is an attention-based weighted average embedding over tokens in the span; $\phi(i)$ is a feature vector to represent the span width feature, which assigns each length with a learnable embedding vector. These vectors are concatenated to represent span $i$.

We use a multi-layer perceptron (MLP) with one hidden layer and ReLU for scoring each span,

$$s(i) = \text{MLP}(\mathbf{g}_i). \quad (2)$$

We rank all possible spans for a sentence in decreasing order of the span scores and keep the top

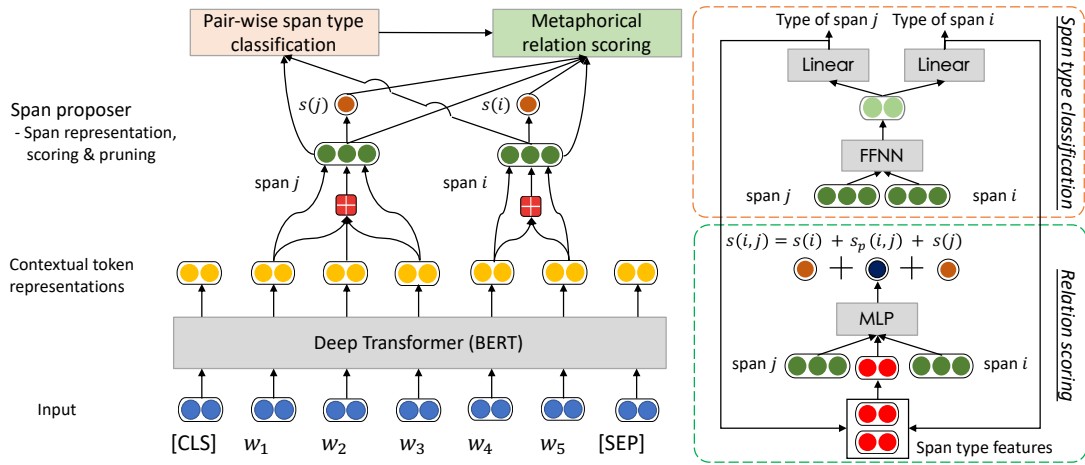

Figure 3: The architecture for the joint model for span type classification and metaphorical relation extraction.

$\lambda$ spans, $0 < \lambda < 1$. We choose $\lambda$ with the development set in experiments. The selected $N$ spans $C_S = [\text{span } 1, ..., \text{span } N]$ would participate in the following stages.

### 4.3 Metaphorical Relation Scoring

A span can form metaphorical relations with multiple other spans. For a pair of spans $(i, j), j < i$, we name span $i$ as the *anchor* and span $j$ as span $i$' *peer*. Our task is to find the correct set of peers $Y(i)$ for every anchor span $i$.

A joint distribution $P(Y_1, ..., Y_N | S)$ over all spans in $C_S$ is formulated as:

$$P(Y_1, ..., Y_N | S) = \prod_{i=1}^{N} P(Y_i | S)$$
$$= \prod_{i=1}^{N} \frac{\sum_{y \in Y(i)} exp(s(i, y))}{\sum_{j \in \mathcal{Y}(i)} exp(s(i, j))}, \quad (3)$$

where $\mathcal{Y}(i) = \{\epsilon, 1, .., i-1\}$ represents all possible assignments, and $\epsilon$ means span $i$ does not have a peer; $s(i, j)$ is the overall score for every candidate pair $(i, j)$. The loss function for relation scoring in sentence $S$ is

$$\mathcal{L}_{rel}(S) = -log P(Y_1, ..., Y_N | S). \quad (4)$$

The scoring function $s(i, j)$ is computed as follows,

$$s(i, j) = \begin{cases} 0, & j = \epsilon; \\ s(i) + s(j) + s_p(i, j), & otherwise, \end{cases} \quad (5)$$

where $s(i)$ and $s(j)$ are the scores computed based on Equation 2 and $s_p(i, j)$ is a pair-wise score for

the span pair,

$$s_p(i, j) = \mathbf{g}_i W \mathbf{g}_j$$
$$+ \text{MLP}([\mathbf{g}_i, \mathbf{g}_j, \mathbf{g}_i \circ \mathbf{g}_j, \mathbf{t}_i, \mathbf{t}_j]), \quad (6)$$

where $W$ is a parameter matrix; $\mathbf{g}_i$ and $\mathbf{g}_j$ are span representations; $\mathbf{g}_i \circ \mathbf{g}_j$ represents the semantic interactions between two spans; $\mathbf{t}_i$ and $\mathbf{t}_j$ are representations of the span type features. The span type features are computed based on the predicted types of span $i$ and span $j$ provided by the pair-wise span type classification module. We will introduce the details later.

### 4.4 Pair-wise Span Type Classification

This module assigns each span in a metaphorical relation a type $e \in \mathcal{E}$. The span types in metaphorical relations are context-sensitive. Consider the following examples, *my heart is broken* and *the city is the heart of US industry*, where *heart* can be used as a *target* or a *source* in different contexts. Therefore, instead of classifying span types independently, we propose a pair-wise span type classification approach.

We feed a pair of spans into a single layer feedforward neural network (FFNN) with `ReLU` as the activation function to get a shared hidden representation and then use two independent linear classifiers to get the probability distributions of the two spans,

$$\hat{\mathbf{y}}_i = \texttt{softmax}(\texttt{Linear}_1(\texttt{FFNN}([\mathbf{g}_i, \mathbf{g}_j]))),$$
$$\hat{\mathbf{y}}_j = \texttt{softmax}(\texttt{Linear}_2(\texttt{FFNN}([\mathbf{g}_i, \mathbf{g}_j]))). \quad (7)$$

We compute the loss function of span type classification across all candidate span pairs that belong to the gold span argument pairs $G(S)$, i.e.,

$$\mathcal{L}_{tp}(S) = \sum_{(i,j) \in R(C_S) \cap G(S)} \Big( \text{CE}(y_i, \hat{\mathbf{y}}_i) \\ + \text{CE}(y_j, \hat{\mathbf{y}}_j) \Big), \tag{8}$$

where $R(C_S) = \{(i,j) | j < i, \text{span } i \in C_S \wedge \text{span } j \in C_S\}$, $y_i, y_j$ are the gold span types, CE is the cross-entropy function. We avoid introducing a *none* type to indicate invalid spans, since this may produce incompatibility predictions, such as a metaphorical relation with a *none* type span.

For computing the span type features in Equation 6, we use a learnable type embedding table $\mathbf{V} \in \mathbb{R}^{d \times |\mathcal{E}|}$ to represent span types, where the dimension is $d$. To reduce the influence of invalid spans and inaccurate predictions during inference, we compute a soft span type representation for span $i$ based on the predicted probability distribution $\hat{\mathbf{y}}_i$,

$$\mathbf{t}_i = \sum_{k=1}^{|\mathcal{E}|} \hat{\mathbf{y}}_{i,k} \cdot \mathbf{V}_k, \tag{9}$$

where $\hat{\mathbf{y}}_{i,k}$ is the probability that the $k$-th type is assigned to span $i$ and $\mathbf{V}_k$ is the representation of the $k$-th type.

### 4.5 Training and Inference

**Training** We combine the loss functions of metaphorical relation scoring and pair-wise span type classification, i.e., $\mathcal{L}_S = \mathcal{L}_{rel}(S) + \mathcal{L}_{tp}(S)$, and train the model sentence by sentence.
**Decoding** Given a sentence for decoding, we first get all span pairs that have a relation score (Equation 5) larger than 0 together with the predicted span types. If multiple pairs share the same anchor and have overlaps between peers, we only keep the one that has the highest relation score. The types of metaphor relations can be inferred by rules based on the types of its spans.

## 5 Experiments

### 5.1 Parameter Settings

We use the Chinese-BERT-WWM-ext model (Cui et al., 2021). For the span proposer, we fix the maximum length $L$ to 15, try $\lambda = 0.4, 0.6, 0.8$, and find that the best performance can be obtained on the development set when $\lambda$ is 0.8 and 92% gold spans are recalled. Concerning the model architecture, we fix the dimension of the span width and span

type features to 20 and fix the embedding dimension of the hidden layers in Equation 2, 6, and 7 to 1000. The span type set $\mathcal{E}$ in span type classification consists of the fine-grained span types.

We tune the hyper-parameters of our model, our variations, and the baseline based on the performance on the development set, and report the results on the test set. We run the experiments with one GeForce RTX 2080Ti GPU. We use the AdamW optimizer with a learning rate of 2e-4 for relation extraction and span type classification.

### 5.2 Comparison Settings

We compare with the following baseline. The baseline follows the working flow of a state-of-the-art relation extraction model (Eberts and Ulges, 2020). It is a joint entity relation extraction model, which classifies the type of each span independently and does not consider span type representations for relation classification.

### 5.3 Evaluating Metrics

We evaluate both metaphorical relation extraction and span extraction with precision (P), recall (R), and F1 as the main evaluation metrics. A span is considered correct if its type and region are both correct. A relation is considered correct if both of its span arguments are correct. We run each model with 3 random seeds and report the average performance.

### 5.4 Comparison Results on Coarse-grained Metaphorical Relation Extraction

### 5.4.1 Overall Results

For comparison with the baseline, we first consider the identification of the coarse-grained span types: target span or source-related span, and a binary metaphorical relation. So this evaluation reflects the ability to identify the existence of metaphorical relations. Table 2 shows the comparison results of relation extraction and span extraction against the baseline. Compared with the joint model baseline, our method has close performance in span extraction but obtains a 1.67% F1 improvement in relation extraction.

Our method adopts the pair-wise span type classification manner and enhances the interaction between span type classification and relation extraction. The results show the effectiveness of the proposed method in identifying metaphorical relations. Each metaphorical relation has two spans. Identifying both span arguments of a metaphorical relation

| | Relation Extraction | | | Span Extraction | | |
|---|---|---|---|---|---|---|
| **Model** | **P** | **R** | **F1** | **P** | **R** | **F1** |
| **Baseline** | 65.77 | 58.64 | 61.79 | 76.13 | 67.90 | 71.62 |
| **Our full model** | **70.71** | 57.59 | **63.46** | **78.99** | 65.57 | **71.64** |
| w/o span type features | 68.25 | 56.42 | 61.67 | 78.55 | 65.04 | 71.08 |
| pair-wise STC →independent STC | 63.50 | **59.82** | 61.34 | 74.13 | **68.99** | 71.29 |

Table 2: Experimental results of the baseline and the proposed model with different settings on metaphorical relation extraction and span extraction.

correctly at the same time is crucial for capturing the meaning and structure of metaphors.

### 5.4.2 Ablation Studies

We further perform ablation studies to analyze the contributions of the pair-wise span type classification module (STC) and the span type features. Table 2 shows the results.

When we remove the span type features, the F1 performance decreases by 1.79% in metaphorical relation extraction and 0.56% in span extraction compared with the full model. Using the span type features benefits metaphorical relation extraction.

When we replace the pair-wise span classification module with an independent span type classification module, the F1 performance decreases by 2.12% in metaphorical relation extraction and 0.35% in span extraction. The pair-wise span classification achieves a large gain in precision for span extraction, which has a significant impact on the prediction of metaphorical relations.

When we remove both modules, our full model degenerates into the one similar to the baseline. By comparison, we can see that the combination of span type features and the pair-wise span type classification keeps a good balance between precision and recall, obtaining the best overall performance.

### 5.5 Analysis on the Identification of Fine-grained Metaphorical Relations

We further analyze the ability to identify fine-grained metaphor relation types, where a span is considered correct if its region and the fine-grained span type are correct. Table 3 shows the results. The simile relation can be identified with the best performance, followed by nominal relation and verb relation identification, while attribute relations are hard to identify.

The results conform to our intuition. The patterns of simile and nominal relations are generally simple, while the patterns related to verb relations

| Model | P | R | F1 |
|---|---|---|---|
| Nominal | 69.03 | 65.55 | 67.24 |
| Verb | 68.04 | 46.81 | 55.46 |
| Attribute | 64.29 | 25.71 | 36.73 |
| Simile | 74.42 | 64.00 | 68.82 |

Table 3: Performance of the proposed method on detecting the fine-grained metaphorical relation types.

are more complex. The attribution relation identification is difficult because of the small number of training examples in our current dataset.

### 5.6 Error Analysis

We analyze the errors in metaphorical relation extraction according to the predictions on the test set. The errors can be summarized into the following types.

(1) Fail to recall (accounting for 60.0%). This type is the most common.

- Sub-type 1: Some errors involve uncommon or abstract concepts as the source, e.g., 自卑心理就像战时的物价高涨起来 (*Inferiority rises like the prices in wartime*), where the source *prices in wartime* also expresses intended meaning.

- Sub-type 2: Some errors are related to sentences with multiple metaphorical relations such as the example in Figure 1(d) and some of these relations are missed. The reasons include: 1) our current model extracts relations independently and does not consider the interaction between multiple relations; 2) in some cases, the long distance between the target and the source increases the difficulty.

- Sub-type 3: Attribute metaphorical relations are often missed. This should be due to the small number of examples in our training data, and our model does not incorporate or learn

concept hierarchies to help capture the object attribute or the whole part relations.

(2) Incorrectly identified relations (34.5%). This type refers to false-positive extracted relations.

- Sub-type 1: Incorrect span region errors are the most common. Some of these errors are acceptable actually. For example, a gold relation is between 温柔、优雅 (*tender and graceful*) and 柳树 (*willows*), while the model may extract the relation between 优雅 (*graceful*) and 柳树 (*willows*).

- Sub-type 2: Some complex syntactic patterns may result in incorrect connections between spans, including the clauses involving the use of the prepositions 把 *(ba)* and 被 *(bei)* constructions, which are two types of non-canonical word orders in Mandarin Chinese. Other error-prone syntactic patterns include appositive and ellipsis.

- Sub-type 3: Some predicted relations involve conventional metaphors, widely used in everyday language and not labeled as metaphorical, such as 束缚-人 (*fetter somebody*), 圆-梦 (*realize a dream*).

(3) Others (5.5%): A few errors are related to the relations that are not annotated during data annotation but seem to be reasonable as well. It indicates that it would be beneficial for data annotation with a machine-in-the-loop manner.

The error analysis can mostly explain the results shown in Table 2 and Table 3. It also reveals that incorporating external knowledge, modeling interactions among multiple relations, and expanding resources for minority metaphorical relation types should be emphasized in the future work.

## 6 Conclusion

We have presented a novel scheme for annotating metaphors in text based on metaphorical relations, which captures the connections between the source and the target through typed spans. Metaphorical relations can reveal the structure and context of metaphors, and facilitate new task formulations for metaphor processing. We create a dataset and develop an end-to-end model for metaphorical relation extraction. We show that our model can achieve moderate performance on metaphorical relation extraction. Our work has opened up a fresh

perspective for metaphor analysis, revealing that metaphorical relations can assume a crucial role in bridging the processing of linguistic metaphors and conceptual metaphors.

## 7 Limitation

Our current work has some limitations that we plan to address in the future.

- First, the size of the dataset can be further expanded, and the distribution imbalance problem among metaphorical relation types should be handled. So the performance on extracting minority relation types can be improved.

- Second, we use metaphorical relations to build surface connections between the target and the source but have not built explicit and complete conceptual mappings, because the sources are often implicit in the text. Therefore we plan to infer complete conceptual mappings based on metaphorical relation extraction in the future work.

- Third, we only use a Chinese dataset for our experiments, but the metaphorical relation schema could be generalized to other languages as well.

## Acknowledgments

This work is supported by the National Natural Science Foundation of China (61876113, 62376166) and the National Key Research and Development Program of China (No. 2022YFC3303504). Wei Song is the corresponding author.

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
