# OpenReview forum: "Chinese Metaphorical Relation Extraction"
_EMNLP/2023/Conference — EMNLP 2023 Findings_

### Official Review · Reviewer_oJ38 · 2023-07-27

**Soundness:** 3

**Excitement:**

3: Ambivalent: It has merits (e.g., it reports state-of-the-art results, the idea is nice), but there are key weaknesses (e.g., it describes incremental work), and it can significantly benefit from another round of revision. However, I won't object to accepting it if my co-reviewers champion it.

**Paper Topic And Main Contributions:**

The paper aims to reformulate the metaphor identification as relation extraction problem, and they created a dataset for Chinese metaphorical relation extraction. Also they propose novel models to this end. The experimental results show that the proposed model achieves the state-of-the-art results in their constructed benchmark, compared to the baseline models.

**Questions For The Authors:**

A, in section 4.3, the scores are not normalized, does the score of the positive pair increase exponentially?

B, in section 4.4, the relation between two spans is classified, there are two losses, one is for the first span and the other is for the second span. Why did you use the two losses for each span, instead of the single loss for relation identification between them?

C, what is the span type feature in equation 9, and how to obtain y_{i,k}.

D, in table 2, span type classification should be after the span extraction. How to use type to span extraction?

E, the span extraction is only <0.7 F1, the results of span type identification should be more lower. It is doubtful that adding the noisy features to the relation classification increase the performance.

**Reasons To Accept:**

1, they created a dataset for Chinese metaphorical relation extraction.

**Reasons To Reject:**

1, the experiments are not solid and some experimental settings is not reasonable, and see question 3 and 4.
2, the experiment results are not convincing, and see question 5.
3, the baseline model is not strong enough.

**Reproducibility:**

3: Could reproduce the results with some difficulty. The settings of parameters are underspecified or subjectively determined; the training/evaluation data are not widely available.

**Reviewer Confidence:**

3: Pretty sure, but there's a chance I missed something. Although I have a good feel for this area in general, I did not carefully check the paper's details, e.g., the math, experimental design, or novelty.

---

> ### Author Rebuttal · Authors · 2023-08-28
>
> Thanks for your comments and questions. We address your concerns here.
>
> > in section 4.3, the scores are not normalized, does the score of the positive pair increase exponentially?
>
> As shown in Eq. 3, we use a product of multinomial distributions for each span. Each term is the probability of a gold pair given the span. Therefore, the scores are actually normalized.
>
> >   in section 4.4, the relation between two spans is classified, there are two losses, one is for the first span and the other is for the second span. Why did you use the two losses for each span, instead of the single loss for relation identification between them?
>
> As we illustrate with an example in Section 4.4, one span can have different types depending on the metaphorical relation it is involved in. This means that we cannot determine the type of each span without considering the pair of spans that form the relation. Moreover, the type of the relation between two spans does not necessarily indicate the type of each span. Our objective is to infer both the span types and the relation types simultaneously. Therefore, we use two losses for each span to optimize their type prediction, and another loss for the relation to optimize its type prediction. In this way, the span type classifier can still receive positive updates even if the relation type is wrongly predicted.
>
> > what is the span type feature in equation 9, and how to obtain $y_{i,k}$.
>
> The span type feature is described in Section 4.3 and is used to calculate the score of each span pair in Eq. 6. The term $y_{i,k​}$ represents the probability of the $k$-th type for the $i$-th span in the sentence, which is obtained from the span type classifier in Eq. 7.
>
> >  in table 2, span type classification should be after the span extraction. How to use type to span extraction?
>
> > the span extraction is only <0.7 F1, the results of span type identification should be more lower. It is doubtful that adding the noisy features to the relation classification increase the performance.
>
> The above concerns relate to the defination of the span extraction task. We define the span extraction task as finding the boundary and type of each span. As we stated in Section 5.2, “a span is considered correct if its type and region are both correct." (match the gold annotation). Therefore, span type identification is included in span extraction. We will make this clear in the revised paper.
>
> > The baseline model is not strong enough.
>
> MelBERT and MrBERT are among the competitive state-of-the-art models for token-level metaphor detection [1][2]. However, our task is different, as we aim to detect metaphorical relations at the span level. To compare our method with them fairly, we evaluate them on the task of metaphorical sentence detection, which is a binary classification problem. In this way, we can measure their performance in identifying sentences that contain metaphorical relations or metaphorical tokens.
>
> [1] Metaphor detection via linguistics enhanced Siamese network, ACL 2022.
> [2] FrameBERT: Conceptual metaphor detection with frame embedding learning, EACL 2023.

---

### Official Review · Reviewer_SXfz · 2023-08-04

**Soundness:** 4

**Excitement:**

4: Strong: This paper deepens the understanding of some phenomenon or lowers the barriers to an existing research direction.

**Missing References:**

Metaphor detection via linguistics enhanced Siamese network - MisNet seems like it is of value to compare to here.

**Paper Topic And Main Contributions:**

This paper presents a method for identification of metaphors by formulating it as a relation extraction process. The paper develops a new dataset that annotates relation extraction according to this paradigm. They present results based on a comparison with a relation extraction system.

**Reasons To Accept:**

This is an interesting and novel view on the metaphor identification process.
This paper develops a new corpus.
The results show a large improvement over state-of-the-art metaphor corpus, although only on the author's corpus

**Reasons To Reject:**

The annotation scheme in 3.1 is very unclear without the examples and even with, I don't fully understand it. It seems that several of the attributes such as attribute seem to be very connected to part-of-speech, even though this is something the authors criticise in other works.
In 3.2.1, The section starting "the annotation process is as follows" is very unclear. How do you define "good consistency and inter-annotator agreement", what exactly is the procedure here?

It would be interesting to see if these results also applied to metaphor corpora used by other authors.

**Reproducibility:**

3: Could reproduce the results with some difficulty. The settings of parameters are underspecified or subjectively determined; the training/evaluation data are not widely available.

**Reviewer Confidence:**

3: Pretty sure, but there's a chance I missed something. Although I have a good feel for this area in general, I did not carefully check the paper's details, e.g., the math, experimental design, or novelty.

---

> ### Author Rebuttal · Authors · 2023-08-28
>
> Thanks for your comments and feedback.
>
> > The annotation scheme in 3.1 is very unclear without the examples and even with, I don't fully understand it. It seems that several of the attributes such as attribute seem to be very connected to part-of-speech, even though this is something the authors criticise in other works.
>
> We will revise the annotation scheme in 3.1. Simply put, our annotation scheme is very similar to an entity relation scheme, where target span and source-related span represent the descriptions related to the metaphorical and literal entities, respectively, and the relation is used to describe the relationship between them. Because the literal entities are often missing in the sentence, the definition of these relations is determined by referring to how the literal entity is manifested in the text, such as the action or attribute of the literal entity. This is indeed similar to part-of-speech, but what we consider here are text spans rather than words.
>
> >  In 3.2.1, The section starting "the annotation process is as follows" is very unclear. How do you define "good consistency and inter-annotator agreement", what exactly is the procedure here?
>
> The annotation procedure consists of three stages. In the first stage, we train the annotators to understand and follow the manual. We emphasize the importance of consistent and reliable annotation, especially on the span boundaries and the existence of the relations. In the second stage, we assign the whole dataset to both annotators and let them annotate independently. In the third stage, we ask them to resolve any conflicts and reach an agreement.
>
> > It would be interesting to see if these results also applied to metaphor corpora used by other authors.
>
> In the future, we plan to annotate metaphorical relations in existing English corpora so that we can compare our approach with previous methods, including the one suggested by the reviewer.

---

### Official Review · Reviewer_7vFy · 2023-08-06

**Soundness:** 4

**Excitement:**

3: Ambivalent: It has merits (e.g., it reports state-of-the-art results, the idea is nice), but there are key weaknesses (e.g., it describes incremental work), and it can significantly benefit from another round of revision. However, I won't object to accepting it if my co-reviewers champion it.

**Paper Topic And Main Contributions:**

This paper focuses on chinese metaphor detection with a focus on characterizing relationships between spans in text that are salient for determining the existence of metaphor and its type. This is different from prior work that either focuses on: a) characterizing/identifying a single word for metphoricity and metaphor prediction is done by using the potentially metaphorical word and its context, b) annotating and identifying syntactic relationsships between pairs of words and using selecitonal preference-based approaches to characterize metaphoricity. This work proposes to annotate spans of contiguous tokens as either pertaining to the “source” domain or the “target” domain that the metaphor maps into. The source spans are characterized to bel;ong to 4 different kinds in the annotation scheme. Based on the source types, 4 types of metaphorical relations are annotated further bertween source spans and target spans. This formalization of metaphor analysis is then used to expand an existing Chinese metaphor dataset with more literal example and finer-grained metaphorical details. Additional filtering is done on the basis of easy-to-predict lexical metaphors and length. A joint span detection and relation type prediction approach is proposed based on BERT representations and is trsained and tested on the annotated dataset. This approach is compared against a pipeline baseline, an existing metaphor detection system, and ablative variants of the proposed approach on span extraction, and metaphor relation/type detection. Analysis on standard binary task of metaphor detection in a single-token + context setting against other baselines is also performed.


**Reasons To Accept:**

– The proposed formalization of metaphor annotation is reasonable and provides a finer-grained perspective of metaphor characterization tasks.

– The dataset production mechanism procedure is soundly described and appears reliable and useful.

– The model seems simple to implement and the loss function/objective is reasonable.

– The proposed approach outperforms pipeline based approaches which shows the importance of joint modeling of span prediction and metaphor relation identification. It also outperforms other approaches on metaphoricity detection task.

– The limitations section identifies the appropriate weaknesses.


**Reasons To Reject:**

– I am not sure if the identified metaphorical relations fully capture the actual rich variation among metaphors. As a non-Chinese speaker, I am not able to comment on it further but I presume that many conceptual metaphors that are not necessarily easily identified by spans of literal text are being ignored by this setting. Similarly, many colloquial usages which show a difference in domains but have become prevalent enough (especially ‘verb’ relation-based metahors) to not be metaphors will be counted as metaphors in this scheme. A further discussion of this will strengthen the paper.

– The dataset appears to be fairly small in size and Figure 2 shows imbalance in terms of metaphor types represented in the dataset. Is this variation natural, or an artifact of annotation guidelines?

– Span enumeration appears expensive in the model and might be difficult to scale to longer text. Moreover, span pruning might help with the cost but it now adds complication to equation 8 which seems to assume that predicted spans include the ground truth spans. Due to discontinuity of pruning, it can make training difficult when generalizing to broader settings. Moreover, it is not clear how the loss in eqn 4 depends on the ground truth. The model description should be improved.

– line 343: I am confused about the description of trigger words. Are these examples excluded or handled separately?


**Reproducibility:**

3: Could reproduce the results with some difficulty. The settings of parameters are underspecified or subjectively determined; the training/evaluation data are not widely available.

**Reviewer Confidence:**

4: Quite sure. I tried to check the important points carefully. It's unlikely, though conceivable, that I missed something that should affect my ratings.

---

> ### Author Rebuttal · Authors · 2023-08-28
>
> Thanks for your insightful comments. We address your concerns here.
>
> > I am not sure if the identified metaphorical relations fully capture the actual rich variation among metaphors. As a non-Chinese speaker, I am not able to comment on it further but I presume that many conceptual metaphors that are not necessarily easily identified by spans of literal text are being ignored by this setting. Similarly, many colloquial usages which show a difference in domains but have become prevalent enough (especially ‘verb’ relation-based metaphors) to not be metaphors will be counted as metaphors in this scheme. A further discussion of this will strengthen the paper.
>
> Metaphors have rich variations, but the metaphorical relations defined here mainly focus on the condition that both source and target domains share in the sentence. They may not be able to capture metonymy, which requires a larger context and common sense to understand.
>
> > The dataset appears to be fairly small in size and Figure 2 shows imbalance in terms of metaphor types represented in the dataset. Is this variation natural, or an artifact of annotation guidelines?
>
> Our dataset is based on a public dataset that has an imbalance issue among the metaphorical relation types. We did not analyze the natural distribution of these types, but it is a good suggestion for future work. We are currently working on alleviating the imbalance issue by incorporating extra data in a semi-supervised way that uses the current classifier to recall more minority types for human annotation.
>
> > Span enumeration appears expensive in the model and might be difficult to scale to longer text. Moreover, span pruning might help with the cost but it now adds complication to equation 8 which seems to assume that predicted spans include the ground truth spans. Due to discontinuity of pruning, it can make training difficult when generalizing to broader settings. Moreover, it is not clear how the loss in eqn 4 depends on the ground truth. The model description should be improved.
> - Currently, our approach is focused on sentence-level processing, so the span enumeration cost is acceptable. The loss function in Eq. 8 is computed over the intersection of the predicted span pairs and the ground truth span pairs, which does not incur additional costs.
> - By optimizing the objective in Eq.3, the model naturally learns to prune spans accurately. Although the initial pruning is random, only gold span pairs can receive positive updates. So the model can quickly make use of such a signal for credit assignment to the different factors, including the span scoring function.
> - Eq. 4 is based on Eq. 3, where $Y_i$​ corresponds to the gold span pairs and $\mathcal{Y}_i$​ denotes all possible span pair predictions. So the connection between Eq. 4 and the ground truth is built.
>
> > line 343: I am confused about the description of trigger words. Are these examples excluded or handled separately?
>
> We use the term “trigger words” to refer to words such as “like” or “as” that are often used in similes. In the original dataset, sentences that contain such words are likely to be simile sentences. This creates a bias that makes the simile relations easy to identify. To avoid this, we add sentences that also contain such words, but are not figurative, to reduce the bias. We will use a more precise terminology and explain this procedure more clearly.

---

### Meta-Review · Area_Chair_Fynp · 2023-09-17

**Recommendation:** 3

**Metareview:**

The authors introduce a new resource for Chinese metaphor relation extraction with almost 10K annotated sentences, consisting of the addition of a new annotation layer to the metaphor dataset by Zhang et al. (2018). They annotate spans of text to identify the source and the target of the metaphor, and they add more fine-grained types for both the source spans (object, action, attribute etc.) and the relations.
Finally, they propose a span-based end-to-end model to extract span pairs from sentences and detect if they are linked by a metaphor relation. Using the newly-annotated dataset, the system is compared with the relation extraction model by Eberts and Ulges (2020) on the span/relation extraction task and with MelBERT/MrBERT on token-level metaphor detection, showing improvements over the baselines in both scenarios.

Among the positives of this paper, the introduction of a new dataset for Chinese metaphor relation extraction and a fine-grained characterization of metaphors in the annotation scheme. On the downside, there is need of some more clarification about the annotation procedure (see discussion with R2) and about the experimental settings and the baseline choice (see discussion with R3).

---

### Decision · Program_Chairs · 2023-10-07

**Decision:**

Accept-Findings

**Comment:**

The authors introduce a new resource for Chinese metaphor relation extraction with almost 10K annotated sentences, consisting of the addition of a new annotation layer to the metaphor dataset by Zhang et al. (2018). They annotate spans of text to identify the source and the target of the metaphor, and they add more fine-grained types for both the source spans (object, action, attribute etc.) and the relations.
Finally, they propose a span-based end-to-end model to extract span pairs from sentences and detect if they are linked by a metaphor relation. Using the newly-annotated dataset, the system is compared with the relation extraction model by Eberts and Ulges (2020) on the span/relation extraction task and with MelBERT/MrBERT on token-level metaphor detection, showing improvements over the baselines in both scenarios.

Among the positives of this paper, the introduction of a new dataset for Chinese metaphor relation extraction and a fine-grained characterization of metaphors in the annotation scheme. On the downside, there is need of some more clarification about the annotation procedure (see discussion with R2) and about the experimental settings and the baseline choice (see discussion with R3).